# Public Dental Service Access Policies for People with Autism Spectrum Disorder (ASD) in Salvador, Bahia, Brazil: A Pre-Evaluation Study

**DOI:** 10.3390/ijerph21050555

**Published:** 2024-04-27

**Authors:** Ednaldo de Jesus-Filho, Sandra Garrido de Barros, Maria Isabel Pereira Vianna, Maria Cristina Teixeira Cangussu

**Affiliations:** 1Family Health Unit of Alto das Pombas, Municipal Health Department of Salvador, Salvador 40226-500, Brazil; edodonto@yahoo.com.br; 2Department of Social and Pediatric Dentistry, Faculty of Dentistry, Federal University of Bahia, Salvador 40110-912, Brazil; isabel@ufba.br (M.I.P.V.); cangussu@ufba.br (M.C.T.C.)

**Keywords:** autism spectrum disorder, dental health services, public health administration, evaluation study, health services accessibility

## Abstract

This study sought to carry out a systematic and preliminary evaluation of the policies on access to public dental services for people with ASD in a Brazilian city. The study, conducted between November/2019 and February/2020, was developed through document analysis, the design of the theoretical logical model of the policies, and seven semi-structured interviews with key informants. The sample was intentionally selected. We also considered the answers to 108 questionnaires from a pilot study on the access of people with ASD to dental services applied to caregivers, dentists, and non-dental professionals. No refusals were recorded. The availability study showed that the policies’ objectives were not being achieved in terms of care network organization: there were no institutional flows, personal contacts were used between professionals to guarantee access to secondary attention, there was no specific training for the dentists about ASD, and the oral health care network was unknown to non-dentist professionals and caregivers. Most people with ASD have visited the dentist at least once in their lives, but a large percentage of those within this study did not do so in the last year. This study identified difficulties in implementing policies and suggested possible strategies for overcoming them as dimensions and subdimensions for evaluation.

## 1. Introduction

People with autism spectrum disorder (ASD) are at greater risk of caries [1]. Tricyclic medications, a diet rich in sucrose, deleterious oral habits, and poor oral health self-care, associated with significant difficulty in accessing preventive dental treatment, contribute to the onset of the disease [2,3,4].

The sensory hypersensitivities associated with autism (light, touch, noise), causing discomfort in face of unfamiliar situations and changes in routine, and difficulty in understanding social interactions and/or in communicating their needs and emotions effectively, make dental treatment difficult [5]. In this sense, dental treatment can trigger stress, physical limitations, aggressive behavior, and voice changes [6].

Access to dental services is generally inadequate, and interventions must be accompanied by actions that make the physical, social, and attitudinal environments more accessible, inclusive, and supportive [7]. The lack of knowledge about the dynamics of the problem and the lack of dentists’ preparation to deal with the specificities of people with ASD, as well as the family’s anxieties and fears, make it difficult to intervene effectively in a timely and successful manner [8].

When caring for people with ASD, it is recommended to maintain silence, avoid too much decoration, and use dimmable lights. Soft ambient music can have a beneficial effect [9]. The behavioral approach to dental treatment should begin with non-pharmacological techniques, which allow for better long-term adherence, especially with children [3,10,11]. However, they require prolonged training times [3]. It is also essential to have a structured schedule and always have the support and participation of the family [9,12].

Treatment should be carried out quickly [9], and picture communication is a promising method [13]. Setting up oral health prevention programs for this population, involving dentists, non-dentists, and caregivers, is essential.

An expansion of access to oral health services in the public health network is a recent achievement in Brazil. In the beginning of this century, Brazilians had difficulty accessing public oral health care services, and the list of procedures was limited [14]. Dental Specialty Centers (CEOs, in Portuguese, Centros de Especialidades Odontológicas) are part of the dental services network as a continuation of the care provided by primary health care (PHC), which is responsible for referring cases [15]. They are implemented through a partnership between states, municipalities, and the Federal Government, which guarantees a financial incentive according to the number of dental chairs (a minimum of three chairs, working 40 h a week). Municipalities complement the resources and sometimes states do too. CEOs must offer, at least, oral diagnostic services, emphasizing the detection of oral cancer, specialized periodontics, minor oral surgery, endodontics, and care for people with disabilities [14,15].

In addition to the rights provided for the Brazilian population in general in the Brazilian Federal Constitution of 1988, in the Statute of the Child and Adolescent (Law No. 8.069/90), and in the Statute of the Elderly (Law No. 10.741/2003), there are specific laws and regulations (Laws No. 7.853/89, 8.742/93, 8.899/94, 10.048/2000, 10.098/2000) [15] for people with disabilities. Since 2011, the federal government has launched specific policies such as the National Plan for People with Disabilities–(Living without Limits), the Care Network for People with Disabilities, the National Policy for the Protection of the Rights of People with Autism Spectrum Disorder (Law No. 12.764/2012) and institutional documents from the Ministry of Health that regulate care for people with ASD in the Unified Health System (SUS, in Portuguese Sistema Único de Saúde) [16].

Although all this legislation and these documents deal specifically with the care for people with disabilities and people with ASD, these are relatively recent and cross-cutting policies, and it is necessary to assess whether they are operating as planned, whether they can be considered to be relatively stable, and what results have already been achieved. This article sought to carry out a systematic and preliminary evaluation of the policy on access to public dental services for people with ASD in the city of Salvador, Bahia, Brazil. Specifically, it aimed to clarify the objectives, responsibilities, resources, and results of the policies under study.

## 2. Materials and Methods

A pre-evaluation study, also known as an evaluability study, was conducted on public policies for oral health care for people with ASD in Salvador, Bahia, Brazil. This type of study consists of a systematic and preliminary examination of the policy or program to identify if it is functioning and may be submitted for an in-depth evaluation [17].

The study was developed through document analysis, the development of the theoretical logical model [18] of the policy under study, and semi-structured interviews with key informants. The sample was intentionally selected. We also utilized the answers to a questionnaire part of the pilot study of an evaluation protocol on the access of people with ASD to dental health services administered at two Children and Youth Psychosocial Care Centers (CAPSis, in Portuguese Centros de Atenção Psicossocial Infanto-Juvenil).

The pre-evaluation study followed four steps: (1) clarifying the objectives of the policy; (2) elaborating the theoretical logical model of the policy; (3) describing how the policy works in practice; and (4) proposing ways of overcoming weaknesses, and dimensions and sub-dimensions for evaluation.

The documents listed in Table 1 were used for the documental analysis. The letter D identifies them throughout the text, followed by the order number in this chart.

The interviews and pilot study were carried out between November 2019 and February 2020, preceding the access evaluation study. No refusals to the interviews or questionnaire applications were recorded.

Interviews and questionnaires were carried out by a single interviewer who had previously been trained. The interviews sought to identify how the programs work in practice, what is known or not known about them, and to map the difficulties in implementing the policies.

The interviews with key informants were scheduled in advance and lasted between 20 and 40 min each. The interview scripts were adjusted according to the position/function of the interviewees (Table 2). The interviews were recorded, transcribed, and analyzed using inductive themes [19]. To guarantee the anonymity of the interviewees, an alphanumeric code was used with the letter I for the interviewees and sequential ordinal numbering (Table 2), removing information that could potentially identify them.

The questionnaires were part of the pilot study of an evaluation protocol on the access of people with ASD to dental health services administered at two Children and Youth Psychosocial Care Centers (CAPSis, in Portuguese Centros de Atenção Psicossocial Infanto-Juvenil). The data were organized into three tables, one for each category of interviewee.

All participants signed an informed consent form. The interview script, a model of the consent form, and a chart with the summary of each analyzed document are available in the Appendix A.

The project was approved by The Faculty of Dentistry of The Federal University of Bahia Ethical Committee.

## 3. Results

Thirteen documents were analyzed (Table 1), seven interviews were conducted (Table 2), and a total of 108 questionnaires were administered in the pilot study (70 to parents and guardians, 14 to dentists, and 24 to non-dental professionals). The results are organized according to the research steps.

### 3.1. Modeling of the Policy

The United Nations (UN) Convention on the Rights of Persons with Disabilities (2006) (D4) reiterated that people with disabilities should have access to all health goods and services without discrimination.

In Brazil, as highlighted in the introduction to this article, people with autism have the rights of the general population provided in the Federal Constitution (D1) and those provided in the Statute of the Child and Adolescent, Law No. 8.069/1990 (D2) and the Statute of the Elderly, Law No. 10.741/2003, as well as rights from specific legislation and regulations to guarantee comprehensive care for these individuals (D7, D8, D9).

In 2011, the National Plan for People with Disabilities–Living without Limits (D5) and the creation, in 2012, of the Care Network for People with Disabilities (RCPD, in Portuguese Rede de Cuidados à Pessoa com Deficiência) (D6) aimed to implement, qualify, and monitor the rehabilitation actions in states and municipalities, providing actions to qualify dental care, in structural terms and staff training from 2011 to 2014 (D8) [20].

Also in 2012, the National Policy for the Protection of the Rights of People with Autism Spectrum Disorders (Law No. 12764/2012), also known as the Berenice Piana Law (D7), began to guarantee a set of actions necessary for comprehensive care for these people [15]. The following year, two institutional documents from the Ministry of Health addressed the care for people with ASD in the Unified Health System (SUS, in Portuguese Sistema Único de Saúde) (D8 and D9).

Health services must operate in an expanded care network, being prepared to welcome and respond to the general health needs of people with ASD, which includes primary and specialized monitoring by habilitation/rehabilitation, medical, dental, and mental health teams, whenever necessary (D8) [20].

The theoretical logical model of the policy (Figure 1) has been constructed from the documents listed in Table 1. The documental analysis revealed two objectives: 1. to involve oral health teams in the oral health care of people with ASD, and 2. to recognize and make non-dental professionals and caregivers aware of the public oral health care network. To achieve these goals, the oral health network needs to involve PHC dentists, specialized dentists, and non-dental professionals; use didactic materials to help with conditioning techniques, such as “Talk-Show-Do” and dental equipment, materials, and instruments; organize the institutional flows between the Coordination of Care for People with Disabilities and the technical area of Oral Health and to define the referral and care protocols for people with ASD between dental services and mental health services.

The program is funded by the Ministry of Health, with co-participation from states and municipalities via Brazilian legislation. The municipalities are responsible for carrying out the services: the dissemination of the referral protocol for oral health specialties and the dental care map for the city among non-dental professionals and caregivers; the training for the oral health team working in PHC to provide dental care for people with ASD, as well as the guidance for caregivers on health care and oral cavity hygiene; and the clinical care for people with ASD.

The expected products are the involvement of PHC dentists, non-dental professionals from the CAPSis and the complementary rehabilitation network, and the study and research groups for the professionals involved; and the inclusion of dentists in rehabilitation environments as a factor in reducing barriers, sensitizing, and encouraging caregivers to condition people with ASD to dental treatment and daily oral hygiene actions.

The expected results are described in detail in Figure 1. Still, in the long-term perspective, they comprise reducing the need for people with mild and moderate ASD to access CEOs and hospitals, a constant dialogue between dentists and non-dentist professionals who treat people with ASD in the municipality, adequate levels of oral hygiene practice by people diagnosed with ASD, the high self-esteem and competence perceived by caregivers, a reduction of barriers to the access and use of public dental services by people with ASD, and the good capacity of the system to deal with the growing number of cases of people with ASD and the aging of this population.

### 3.2. The Policies on Access to Dental Services for People with ASD: Clarifying Their Objectives

In the municipality under study, people with ASD should enter the public health system through PHC, which can refer them to Dental Specialty Centers (CEOs) as necessary, which happens most of the time, primarily through the Center for Attention to People with Disabilities (NAPES, in Portuguese, Núcleo de Atenção a Pacientes Especiais) (D10, D13). These pathways to accessing the oral health care networks are in the interviewees’ speech. However, the availability study showed that the policies’ objectives were not being achieved in terms of the care network’s organization: there were no institutional flows, personal contacts were used between professionals to guarantee access to secondary attention, there was no specific training for the dentists about ASD, and the oral health care network was unknown to non-dentist professionals. When the data of the pilot study were analyzed, the majority of the people with ASD had visited the dentist at least once in their lives, but a large percentage of them did not do so in the last year.

The following sections present the main problems identified in the evaluability analysis and the main dimensions and sub-dimensions for evaluation.

### 3.3. How the Policy Works in Practice

#### Flows, Protocols, and Continuing Education: Coordination Strategies

No specific institutional flow was identified between the people with disabilities coordination and the technical area of oral health, or between the care networks for people with disabilities, or the psychosocial care network, or between these and the health units providing dental care. The link between the oral health services and the care network for people with disabilities in the municipality occurred through personal contact between the professionals.

[…] There is no institutional flow between these coordinations. […] there is no direct relationship with oral health coordination […].(I1)

[…] The CEOs themselves contact the CAPS, looking for patients with a profile for NAPES, not just those with ASD since many users use the CAPS as a gateway instead of primary health care equipment.(I2)

[…] The functioning of the network is still fragile. Many things are achieved not through formal, institutional means but through personal contact […].(I5)

There was also a need to establish protocols and strengthen matrix support in health care.

[…] During Activities of Daily Living (ADL), there should be time for oral hygiene education. […] A psychologist or occupational therapist cannot always carry out oral health education.(I7)

[…] the professional responsible for the care, when called upon […], goes to this PHC Unit to discuss the case with the professional from that health unit or accompany the user to be cared for, as they already have the link created with the CAPS professional.(I1)

Matrix support is highlighted as a positive strategy, but it does not occur systematically and needs to be better adhered to by PHC professionals.

There is no specific theme for the dental treatment of people with ASD. […] The strategy would be to have a dialog with the professionals at the CEOs so that they can point out the obstacles to conducting the process as a whole.(E2-SB coordinator)

There has not yet been any matrix support for people with ASD. […].(E3-DSC manager)

Concerning primary care, there has already been an opportunity for matrix support, but adherence is very low […](E4-CD NAPES)

The national policy seeks to induce this articulation with the care network for people with disabilities by setting targets and linking them to the financial resources transferred to municipalities and states. However, dental care for people with ASD is planned to be provided at CEOs that have NAPES and not at PHC locations (D3).

Emergency care was seen as a shortcut in access to specialized care. After diagnosing the oral health condition of the person with ASD, an informal flow is established, enabling access and the continuity of care after the first visit. Access also occurred through hospitals or rehabilitation centers, although there was a consensus among those interviewed that the main gateway to public dental care should be PHC units.

[…] There are reports of family members preferring to wait for a time of urgency and emergency before taking action about oral health conditions, […](I1)

The system adopted by the municipality to record PHC actions consolidated reports that do not allow for the cross-referencing of variables or the identification of people with ASD living in the city.

Generally, training is aimed at people with disabilities and does not delve into specific conditions such as ASD. There is a need to train and sensitize the oral health team, especially the auxiliary staff, to care for people with ASD.

[…] There has been training for people with disabilities in general, not specifically for those with ASD. […].(I2)

In addition to the barriers familiar to all SUS users (a lack of vacancies, lack of human resources and/or materials, inadequate or insufficient professional training to care for people with disabilities, geographical distance), people with ASD also face specific situations that hinder their access to both medical and dental health services. Some behaviors of people with ASD can be an essential barrier to dental care.

[…] The person with ASD needs to be won over; a bond needs to be created between the professional and the autistic person. Health professionals often do not have time for this process. The demand from children who need dental treatment is very high.(I7)

People with ASD are referred directly by the CAPS or PHC to a CEO. The professionals tailor the service to each person’s needs, reducing waiting times and keeping the same oral health team to establish a bond. The limits of the physical and material structure, even in the CEOs, do not allow for more complex procedures, such as minor oral surgery, endodontics, specialized periodontics, and prosthetics, and there is a need to adapt the spaces for greater accessibility for people with physical disabilities.

[…] attention is paid to the time the person with the disorder is seen so that waiting times are kept to a minimum […] it is requested that harmony is maintained within the room, avoiding unnecessary movement in and out […]. We also try to keep the same professional assisting each patient so that a certain bond can be created.(I3)

[…] One positive thing we’ve achieved is the possibility of having a permanent assistant to care for people with ASD at NAPES, which helps a lot in forming a bond. […] What really stands out is the need for more structure for complex care.(I4)

Caregivers and health professionals have differing views on the presence of a dentist’s office in rehabilitation centers.

The presence of a dentist in institutions that regularly care for people with ASD would enable work on oral health to be carried out […] the environment in which the person with ASD is already inserted would be a way of creating this bond and making care feasible. […] The dental office doesn’t play this role of social inclusion of the person with ASD.(I7)

[…] The user shouldn’t have to stay inside the facility to meet all their demands. […] There should be intersectorality and the capillarity of these people throughout the network, not considering what they can find outside the CAPS. […](I1)

[…] the movement made by mental health professionals is not to “isolate” people with disabilities. […] People with special needs need to circulate in other spaces. […] (I5)

Parents feel safer in specialized services and often delay dental care until the age of 7 to 14, when the majority of services do not accept treatment.

[…] The process of acceptance is slower and takes longer for some, which delays the movement towards oral health care as well […] (I7)

[…] Some parents say they prefer not to be public with their children […]. Many families avoid taking people with ASD to the health center because they don’t believe they’ll find a qualified professional there.(I1)

[…] some parents don’t want their children to be seen by a professional who isn’t specialized. […] They already bring the demand for hospital-level care under general anesthesia. It turns out that many are surprised by their own children with ASD when they realize that the PHC dentist has managed to perform supervised brushing, for example […] (I6)

The lack of information about oral health care for people with ASD and patients with special needs in general in the SUS often leads to a surplus of vacancies. At the same time, caregivers complain about the lack of vacancies.

[…] There is a great deal of ignorance about the dental service offered by the municipality to people with special needs through the CEOs since there is a great deal of complaining about the lack of vacancies by caregivers, while there are vacancies available in the network to be filled.(I3)

### 3.4. Relationship between Primary, Secondary, and Tertiary Care

The importance of PHC in building bonds with users and their families is highlighted. Still, the interviewees point to the lack of structure in PHC for caring for people with disabilities, which is also related to inadequate professional training.

[…] PHC still can’t see that people with disabilities live in that community, that they are part of that territory, which is a dynamic territory, and these people aren’t cared for.(I1)

[…] PHC plays a fundamental role in caring for people with ASD, mainly because of the bond created with the family. In addition, the specialist at the CEO needs more time to create this affinity. […](I2)

There are delays in secondary and tertiary care referrals, as has already been mentioned regarding scheduling the first appointment. The municipality’s recommendation to treat people with ASD in the “red rooms” of the Emergency Care Units, since they are prepared for conscious sedation, reinforces the access to services through emergency services.

[…] The health center refers them to the CEO. At the CEO, there are already difficulties in providing care… The delay in referrals from the CEO to the hospitals is even longer.(I7)

Unfortunately, hospital care for patients referred by this center is not working. Patients find it very difficult, even before the pandemic. […](I4)

The oral health coordinator points out a strategy of direct referral from PHC to hospital care; a dentist from the DSC confirms that the flow should be direct, while a CEO manager points out that there is misinformation.

[…] The town hall has created an agreement with a hospital in the city that receives patients referred directly from PHC in an attempt to reduce the referral time between services. […] the city’s Emergency Care Units have the so-called “Red Room”, with all the technical equipment to carry out conscious sedation, and it is suggested that, if necessary, this procedure be carried out in these municipal facilities.(I2)

The person with special needs is referred via the Referral and Counter-Referral Form. Due to a lack of information, the CEO receives several referrals indicating sedation. So, the user (or their guardian) is instructed at the CEO’s reception desk to present this same form to the hospitals that perform this type of procedure. (I3)

The importance of the referral and counter-referral system (RCRS) for better health care for people with ASD is emphasized, although, how it works and the importance of informal contact involving professionals in the process are not pointed out.

[…] Every referral made by the Referral Form must have a counter-referral from the professional looking after this person […]. Sometimes, telephone and informal contact between services, beyond any form, for example, is potent. […] This is an “implicated” referral when the patient’s origin and destination are directly connected, from professional to professional, from service to service.(I1)

Another issue pointed out is the lack of clarity among the interviewees about the concept of people with special needs, which conditions are included, and whether people with ASD are classified as people with disabilities.

[…] there needs to be more understanding about what a person with special needs is. Hypertensive patients, people with diabetes, and pregnant women are sometimes mistakenly classified as people with special needs. As a result, the CEO does not accept some referrals because the person doesn’t fit the NAPES prerequisites. (I3)

There are various programs and thematic fields that the professionals at the town hall have to carry out. None are specific to people with ASD. Perhaps they include people with disabilities! But autistic people are not disabled people! I think this is a necessary and urgent discussion […].(E6-CD USF)

### 3.5. Provision of and Access to Dental Services by People with ASD: Results of a Pilot Study

The questionnaires were administered to parents and guardians at two CAPSis, and no refusals were registered. The number of participants, based on the users and professionals in the two services, was determined on the day of the pilot study’s application. The results showed that most (72.46%) of the people with ASD had already visited the dentist, but 57.14% had not done so in the last year. The caregivers (65.22%) were unaware of the places for dental care provided by the SUS. They encountered difficulties in obtaining dental care (69.12%), with scheduling appointments (52.73%) being the main problem mentioned, and the majority (81.82%) said they had already received oral health advice (Table 3).

The non-dental professionals had backgrounds in various areas and levels of training (administration, social work, nursing, pharmacy, medicine, psychology, nursing technician, occupational therapy, among others). Most of them reported not practicing supervised oral hygiene (83.33%), not receiving materials (62.50%), and not having a suitable place to do it (83.33%) in the institution where they worked. There was also no distribution of oral hygiene kits (87.50%), and caregivers/guardians were not instructed on the oral hygiene of people with ASD (83.33%), nor were regular visits by dental professionals (70.83%) or activities by undergraduate dental courses (82.61%) reported in these units. When they did happen, it was only occasionally. The professionals reported that they did not know the flow of dental care for people with ASD, either in the municipal (91.67%) or in the state (100.00%) public oral health care network, nor did they know where the users seen at the CAPSis received dental treatment (66.67%). If dental care was needed, referrals were made to PHC units (66.67%). There were no dental practices at the CAPSis, and non-dental professionals felt that dental treatment for people with ASD should take place in the same environment as people without ASD (79.17%) (Table 4).

The dentists interviewed felt that there was no need for differentiated care for people with ASD compared to other patients with special needs (57.14%). In this sense, there were no differences in scheduling appointments (57.14%), specific training for dentists to care for people with ASD (57.14%), places reserved for this group (92.86%), or specific oral health programs (85.71%). Most dentists were not qualified to use nitrous oxide (71.43%), and no operating room was available to treat people with ASD (78.57%) in their workplaces. According to the dentists, it takes around a week for the first appointment to be scheduled (46.15%), although 30.77% did not know this, and said that all guardians of people with ASD were informed about the oral health of people with ASD (Table 5).

### 3.6. Ways of Overcoming Weakness and Dimensions and Subdimensions for Evaluation

The municipality’s administration should improve care coordination by formulating specific institutional protocols and flows between the people with disabilities and oral health organizations and between the care network for people with disabilities or the psychosocial care network and the oral health care units. The matrix support between NAPES and PHC professionals must occur systematically, and PHC dentists should be better informed of how they can demand it and how it works. This strategy of permanent education should be complemented by the training of the oral health teams, not only for disabilities in general but also focusing on some specific conditions such as ASD.

The PHC team must be able to welcome, diagnose, treat, and, when necessary, refer people with ASD to secondary or tertiary care. For this purpose, it is also necessary to establish protocols of reference and counter-reference to procedures and levels of ASD that can be carried out in PHC units, specialty centers, and hospitals.

It is also essential to define ASD as a disability and recognize its specificities and different levels and implications for oral health care.

The dental health care network for people with ASD must be known by all professionals involved in providing care for people with ASD.

Based on the aspects analyzed through the strategies adopted in this study, three main dimensions were identified for evaluation: management, health care, and monitoring and evaluation. The assessment should address the different components of these dimensions (their structures, processes, and results) [21], as well as the impacts of implementing the policies (the early diagnosis, prevention, and stabilization of oral health conditions, and improving the quality of life of people with ASD, and of caregivers and family members) (Figure 2).

## 4. Discussion

In general, health policies have made significant progress over the years in terms of guaranteeing access to health care for people with all types of disabilities and mental disorders, structuring a care network that meets their diverse demands, and respecting their uniqueness. Public policies are based on the logic of bringing these individuals back to social life, with treatments integrated into the community, organized in a network and an intersectoral manner, guaranteeing rights of access to the job market and rehabilitation technologies [22]. However, this is yet to be a reality in oral health care in the municipality, as observed in this study’s results.

People with ASD have similar oral pathologies to people without ASD. Thus, there are no significant differences in care compared to the general population. The main problem is precisely their cooperation and behavior at dental appointments due to the specificities of their condition [23].

Although there is no direct relationship between autism and oral diseases, individuals with ASD may have worse oral health conditions than those found in the general population [23,24,25]. They are at a greater risk of developing oral diseases, such as caries and periodontal disease, and early diagnosis is essential to provide more appropriate therapeutic approaches [26].

The early diagnosis of ASD and associated oral health problems helps to develop social, communicational, and behavioral approaches at an early age, having a positive effect, for example, on eliminating bruxism and its consequences [27].

More significant oral health needs imply a greater demand for oral health services [28], determining their use [29]. Individuals with greater vulnerability may be more affected by oral health morbidities. People with ASD have a high frequency of unresolved dental needs, reinforced by a lack of demand for dental treatment [6] and more restricted access to oral health services [30,31]. In this sense, specific public policies for people with ASD to access dental services are essential.

In Salvador, there was a lack of knowledge among caregivers, dentists, and non-dental professionals about the oral health care network, as well as the relationship between ASD and oral health. The unpreparedness of the PHC oral health network and the low use of specialized dental services were also noted.

The lack of information about dental services’ supply, availability, and importance can hinder their use [32]. Professionals must be familiar with the health care network to direct these people through the RCRS, aiming for comprehensive care and multi-professional action. Public policies and coping strategies at different levels of health care are essential aspects to consider. Families and professionals, both dentists and non-dentists, need to develop strategies to deal with these situations and help the individual with ASD to feel safe and comfortable [33], a situation that is particularly important in dental care, as seen in this study, and which most professionals, dentists, and non-dentists, are unaware of.

Public policies aimed at this specific population must guarantee interdisciplinary care and access to the entire health care network for people with disabilities. Mapping and publicizing the services aimed at these people is fundamental for more timely access to health services. A lack of adequate publicity about services can lead to their low utilization due to a lack of knowledge, even when there is demand, as observed in this study.

Access to support services for people with ASD is often inadequate [34,35]. Even though the SUS is guided by constitutional principles that guarantee universal access to health services, the most vulnerable populations still face inequalities in the access and use of services [36,37]. Despite the expansion of primary oral health care coverage in recent decades in Brazil [38], the scheduling of the first dental appointment was one of the problems highlighted by caregivers for the dental care for people with ASD.

The Brazilian Ministry of Health recommends that dental care should take place primarily within the scope of PHC, and that there should be specialized and hospital referral units for more complex cases and those requiring care under general anesthesia [39]. The insufficient training of PHC dentists in caring for people with ASD leads to a rejection of this population, which is excessively referred to specialized care. It should also be noted that PHC has more units with dental services and greater capillarity throughout the municipality.

Emergency services are also a gateway, but they should not be the main form of access to the health system, which has been the case for people with ASD in this municipality. This has been configured as a strategy for more timely access to specialized services, which takes place through personal contacts between professionals rather than through pre-established institutional flows, which generate inequalities in access.

The principles of PHC and, in particular, the Family Health Strategy (ESF, in Portuguese, Estratégia de Saúde da Familia), advocate client adscription, registration, and family monitoring so that, at the very least, these individuals should be included in the list of people with special needs in the area covered and monitored every month. This would also help dentists to actively seek them out, establish a link with the PHC team, and carry out home visits and care when indicated.

According to the National Primary Care Policy [40], it is part of the work process for PHC teams to provide health care at home. Home visits can be a strategy to reduce the distance between health units and people with ASD.

Public health programs and policies such as the Family Health Strategy (ESF), the National Primary Care Policy, and the National Oral Health Policy were created to bring public health services closer to the principles and guidelines of the SUS [41]. The everyday duties of all PHC professionals include providing humanized care, taking responsibility for the continuity of care, and making it possible to establish a bond [40]. To increase the frequency of children’s early use of oral health services, greater attention should be paid to parents who do not use them regularly [28]. The offer of dental care on weekends could facilitate the use of services and the return of parents with their children, as the dentist would be available during shifts and days when caregivers are usually not working [42].

No survey of the oral health conditions of autistic people was identified in the city of Salvador. Studies of this kind are rare, even among institutionalized children with ASD [43]. It is worth noting that in addition to knowing the oral health conditions of this population, the information obtained can support the planning and organization of actions and services.

To provide dental care for patients with ASD, well-defined protocols must be followed [44]. The difficulty of dental treatment is described in some studies, but the adaptations to make dental treatment and oral hygiene possible are hardly found in the literature [45,46,47]. People with ASD adapt and collaborate better when the same team, dentist, and clinical environment are maintained at each appointment. It is also essential that the person responsible remains in the office [48,49]. These precautions must be observed at all levels of health care.

One of the alternatives for dealing with the challenges posed by caring for people with ASD is to take part in parent and professional training programs offering guidance and strategies to help families deal with the difficulties of ASD, which can include information on communication, behavior, and social skills, as well as counseling and emotional support, elements that were lacking at all levels of care, as well as the availability of supplies to meet this demand [50].

In addition to non-dental professionals, health education programs should also target caregivers so that everyone is informed about the importance of oral hygiene practices. The lack of knowledge about oral health and the little importance given to primary teeth, as well as the fears and myths created by the caregivers themselves about dental treatment, create additional barriers to preventive and early treatment for people with ASD [51].

The interaction with other health professionals makes it easier for dentists to get closer to people with ASD. The role of the pediatrician is fundamental for the oral health of autistic children. Intervention and referral to a specialist can be made early in life, increasing the acceptance of treatment and allowing a bond to be forged early on with the dentist. The dentist can take the correct approach and ensure prophylactic therapies and adequate care [52]. Another possibility is matrix support, a strategy of permanent education through dialog between professionals, in this case, professionals from CAPSs, CEOs, and PHC.

Within the scope of the SUS, especially considering the principle of comprehensiveness, the legislation ensures that autistic individuals can meet their demands in their entirety in terms of their biological, psychological, and sociocultural aspects [53]. As this is a recent policy, evaluative studies can help improve it. Insufficient access to public health services and social support increases the emotional burden on parents [54], and in the medium term, improving the policy would also contribute to reducing stressors among family members and caregivers.

When treatment is unsuccessful in the office, dental treatment should be carried out using the induction of general anesthesia, making it possible to carry out total oral rehabilitation in a single session, from prophylaxis to surgery [55]. However, most dentists were not qualified to use nitrous oxide if they needed sedation. We highlight that this procedure should be done outside of PHC in Brazil.

Continuing education for dental professionals and caregivers is essential to overcome the difficulties encountered by people with ASD in the dental chair [24]. The dental treatment of a child with ASD requires an understanding of the behavioral profile of the person with autism [24].

Barriers to dental care have been cited by caregivers, which can influence their access to regular care and the interval between appointments [56,57,58]. The waiting time for scheduled appointments at public dental services varied in some studies from 15 days [59] to 30 days or more [60]. There was disagreement between the answers given by caregivers and dental professionals regarding the waiting time for the first dental appointment.

Accessibility is greatly influenced by the organizational characteristics of the units, which reflect the different professional profiles and local management [61]. In the ESF, with a few exceptions, dentists do not work at night or on weekends, making access difficult for children dependent on parents who work during the day [62]. However, there is only a point in having a third shift if the service is appropriately qualified and publicized to the population.

According to Paim (2003) [17], the way services are organized, based on the organized supply of actions, tends to overcome the traditional ways of reorganizing the production of health actions. If, on the one hand, some of them help to program the supply of units, on the other, they generate frustration among users who want the opportunities given to some groups to be offered to everyone [63].

Children and adolescents followed up with by the family health team are more likely to visit the dentist, demonstrating the longitudinality of care and a bond between users and the team [64,65]. These aspects must be considered when drawing up public oral health policies. Equity as a perspective of expanding access must be considered in constructing the suggested policies [28]. Equity in health increases social justice in access to goods and services [66]. Identifying groups vulnerable to risks and who may not access health services adequately would be fundamental to the scientific and political approach to health inequalities. In this sense, it is worth reflecting on the priority that can be given to people with ASD for oral health care.

It is still challenging for dentists to work in the ESF due to their conceptions of the strategy and the autonomy it grants them, and most of them restrict their work to practicing in the dental office [64]. To fulfill the duties of the family health teams designed by the Ministry of Health [67], oral health professionals need to have competencies and skills that are initially built up during their training and constantly improved during the process of continuing education to promote health, prevent illness, and make clinical procedures, working considering the social determinants of health, in a multidisciplinary team with an interdisciplinary actuation [68].

It is also known that the treatment of ASD depends strictly on its diagnosis and the degrees of support. In this sense, the various interventions, such as behavioral therapies, educational interventions, occupational therapies, and medication, should be decided together with the multidisciplinary team, considering each child’s individual needs [69]. Thus, some children could be monitored by PHC, while others should receive continuous and personalized treatment closely monitored by the professionals involved [70]. There is still little clarity about the specific needs of people with ASD in the municipal care network.

Non-dental professionals referred autistic people to health services, either PHC units or CEOs, when they needed dental care. The CEOs may represent an additional attraction for the service because they can solve more complex health needs, such as caring for people with special needs [16]. It should be noted, however, that it is up to someone other than non-dental professionals to refer people to specialized services.

Research on dental care for people with ASD has gained prominence in recent years [52,71,72,73,74,75], and this study attempts to contribute to a greater understanding of the public policies for access to oral health care for people with ASD in the city studied.

The involvement of non-dental professionals was fundamental. Knowledge of oral health conditions, access to, and the use of the dental services offered by the SUS is as crucial as it is unknown to this category, and it is clear that one of the significant barriers is the lack of information.

The progress of Brazilian health policy is a concrete fact, consolidated by the measures gradually implemented to provide well-being and comprehensive, humanized care for people with ASD. This fact becomes even more critical when we look at the concern for policies that specifically meet the demands of people with ASD. The Ministry of Health’s efforts to organize a care network and offer specialized rehabilitation services are of great importance and are consistent with the principles of the SUS. Thus, it is essential to highlight and acknowledge these advances and efforts, but also to recognize the need to define better the roles of the institutions involved in specialized care and the (evidence-based) rehabilitation services that will be offered by qualified professionals for their practice [22], as well as to invest in the peculiarities of the care networks that demand specificities for this population, such as oral health.

The study’s limitations include that only one representative from each category was heard and that they were not chosen randomly. Due to the numerous peculiarities inherent in the condition of ASD itself and the different worldviews of each interviewee, it would be necessary to extend the research to other families and other health professionals.

As a contribution to this study, the dentist’s role in caring for people with ASD must be discussed in the training of future professionals and in ongoing education to enable the access, reception, and monitoring of these people within the health network. To this end, PHC should be valued and optimized, focusing on health promotion and disease prevention, especially for children with particular health demands.

It is hoped that studies such as this one will help to draw up specific public policies for people with ASD, taking into account their specificities and unique demands for multidisciplinary and comprehensive health care.

## 5. Conclusions

Although Brazilian public health policies guarantee the access of ASD people to public dental services, this right has not been entirely improved in the city under study here. This study identified difficulties in implementing policies and suggested possible strategies for overcoming those difficulties as dimensions and subdimensions for evaluating the oral health care policies for people with ASD in Brazil.

Given the findings, some considerations and recommendations can be formulated to help managers improve the strategies analyzed here: (1) The evaluability study showed that in the city of Salvador, although the health administration pretends to implement the policy of access for people with ASD to dental services, there are coordination, structural, and personnel barriers to overcome; (2) There is an unmet demand for public dental services by people diagnosed with ASD in the city under study, which requires the reformulation or the improvement of the strategies to respond to this need; (3) There is a possibility of increasing the accessibility of public dental services for people with ASD by reviewing the programs.

There was a lack of understanding among the professionals working in the municipality’s PHC of their role in this process. Contrary to the hegemonic model, which favors super-specialized professionals, the family dentist must act as a generalist, covering all life cycles and health conditions. Few professionals understand PHC as a space for caring for people with ASD. Since PHC has the role of coordinating care, its professionals must not only promote the inclusion of people with ASD in the health care network but also monitor their trajectory within it.

The PHC dentist’s role does not involve referring people with ASD to specialized care. They must coordinate care. This can only be achieved with a counter-referral from specialized care, informing PHC about the care provided to that user. These processes, however, have yet to be guided by municipal administration.

Actions need to be planned together with the multi-professional team, and there is no space for the dentist to be isolated within the office. It is also worth pointing out that each service has its own model and policy, meaning each health network has a specific demand. Due to the organization of its residents and the epidemiological demands in the municipality in question, it was established that the CAPS team should consist of a nursing technician, a physiotherapist, an occupational therapist, a psychologist, a clinical and a psychiatric doctor, a physical educator, a nurse, and a social worker. However, we must consider the possibility that the municipality sees the presence of a professional on the team as necessary, as is also pointed out in the literature and by the demands of this population.

## Figures and Tables

**Figure 1 ijerph-21-00555-f001:**
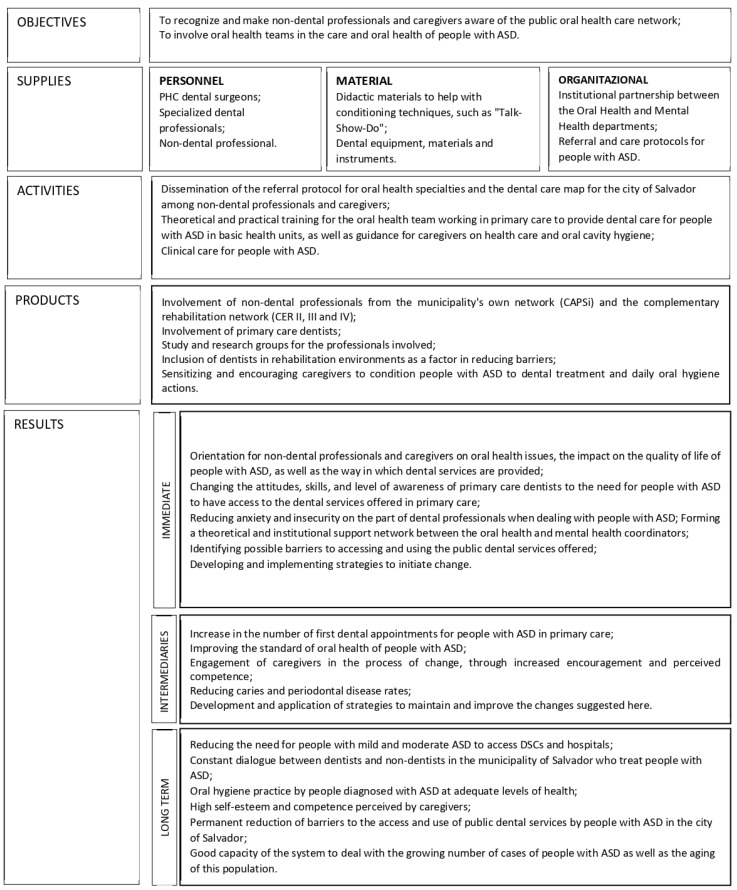
Theoretical logical model of the access policy for people with ASD to public dental services in a capital city in northeastern Brazil, 2020.

**Figure 2 ijerph-21-00555-f002:**
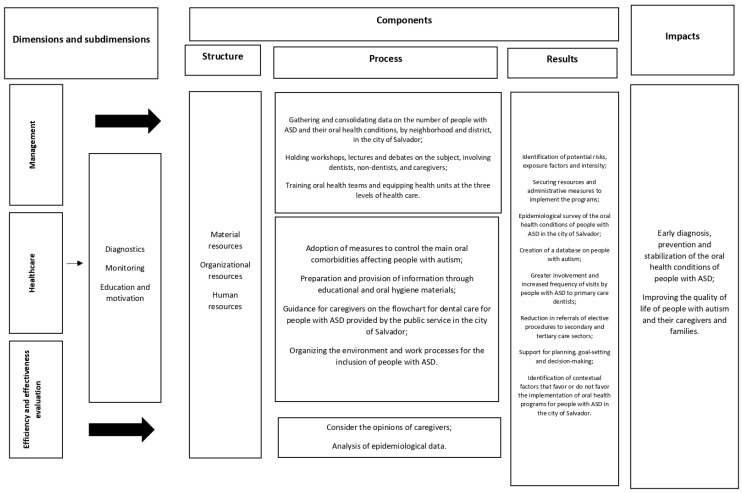
The dimensions and sub-dimensions for evaluation and the criteria related to their development (structures, processes, and results).

**Table 1 ijerph-21-00555-t001:** List of documents analyzed to create the Logical Theoretical Model for evaluating oral health programs for people with ASD in Salvador, Bahia, Brazil, 2019–2020.

N	Document Title	Publication Year	Source
D1	The Brazilian Federal Constitution	1988	Brazilian Federal Government
D2	The Child and Adolescent Statute	1990	Brazilian Federal Government
D3	National Oral Health Policy	2004	Brazilian Federal Government
D4	Convention on the Rights of Persons with Disabilities	2006	United Nations (UN)
D5	National Plan for People with Disabilities–Living without Limits	2011	Brazilian Federal Government
D6	Care Network for People with Disabilities	2012	Brazilian Federal Government
D7	National Policy for the Protection of the Rights of People with Autism Spectrum Disorders (Berenice Piana Law)	2012	Brazilian Federal Government
D8	Guidelines for the Rehabilitation of People with Autism Spectrum Disorders (ASD)	2014	Brazilian Federal Government
D9	Care Pathway for People with Autism Spectrum Disorders and their Families in the Psychosocial Care Network of the Unified Health System	2016	Brazilian Federal Government
D10	Referral Protocol for Oral Health Specialties in Salvador	2017	Salvador City Hall
D11	Oral Health in the Unified Health System	2018	Brazilian Federal Government
D12	Guide to Oral Health Care for People with Disabilities 2019	2019	Brazilian Federal Government
D13	The Municipal Health Plan 2018–2021	2018	Salvador City Hall

**Table 2 ijerph-21-00555-t002:** List of interviewees for the elaboration of the Logical Theoretical Model for the evaluation of oral health programs for people with ASD in a Brazilian Municipality, 2019–2020.

N	Interviewees
I1	Representative of the Coordination of Care for People with Disabilities of the Municipal Health Secretariat of the City of Salvador
I2	Representative of the technical area of Oral Health of the Municipal Health Secretariat of the city of Salvador
I3	Dental Specialty Center Manager
I4	Dental surgeon at the Center for Attention to People with Disabilities
I5	District dentist
I6	Dentist working in the Family Health Strategy
I7	Caregiver of a person with ASD

**Table 3 ijerph-21-00555-t003:** Considerations on the oral health of people with ASD based on information from their caregivers, 2019–2020.

Variables	n	%
Has the person with ASD ever been to the dentist?		
Yes	50	72.46
No	18	26.09
Didn’t know	1	1.45
Has the person with ASD been to the dentist in the last year?		
Yes	30	42.86
No	40	57.14
Does the person responsible for the person with ASD know of a place for dental care for ASD through the SUS?		
Yes	22	31.88
No	45	65.22
Didn’t know	2	2.90
Does the person with ASD find dental care difficult?		
Yes	47	69.12
No	14	20.59
Didn’t know	7	10.29
What kind of difficulties do people with ASD have in accessing dental treatment?		
Can’t get an appointment	29	52.73
The service location is far from home	5	9.09
Difficulty obtaining referral forms	8	14.55
More than one situation	11	20.00
Never looked for or had no difficulties	2	3.64
Has the person responsible for the person with ASD received oral hygiene advice?		
Yes	54	81.82
No	11	16.67
Didn’t know	1	1.52

**Table 4 ijerph-21-00555-t004:** Information from non-dental professionals on oral health behaviors and actions aimed at people with ASD, 2019–2020.

Variables	n	%
Place of research		
CAPSi 1	11	45.83
CAPSi 2	13	54.17
Is there an oral health program in the institution?		
Yes	1	4.17
No	22	91.67
Didn’t know	1	4.17
Is oral hygiene practiced in the institution?		
Yes	2	8.33
No	20	83.33
Didn’t know	2	8.33
Does the institution receive material for oral hygiene practice?		
Yes	6	25.00
No	15	62.50
Didn’t know	3	12.25
Is there a place in the institution for practicing oral hygiene?		
Yes	3	12.50
No	20	83.33
Didn’t know	1	4.17
Does the institution distribute oral hygiene materials?		
Yes	1	4.17
No	21	87.50
Didn’t know	2	8.33
Do people with ASD and their guardians receive oral hygiene advice?		
Yes	3	12.50
No	20	83.33
Didn’t know	1	4.17
Does the institution receive visits from dental professionals?		
Yes	5	20.83
No	17	70.83
Didn’t know	2	8.33
If yes, how often?		
Annually	2	66.67
On request	1	33.3
Does the institution receive visits from dental schools?		
Yes	2	8.70
No	19	82.61
Didn’t know/Didn’t answer	3	12.50
Do you know the Municipal Health Secretariat flowchart for dental care for people with ASD?		
Yes	2	8.33
No	22	91.67
Do you know the State Health Secretariat flowchart for dental care for people with ASD?		
No	24	100.00
Do people with ASD in your institution receive dental treatment somewhere?		
Yes	4	16.67
No	4	16.67
Didn’t know	16	66.67
If the patient needs dental care, does the institution refer them for treatment?		
Yes	16	66.67
No	4	16.67
Didn’t know	4	16.67
Which dental treatment facility does the institution refer you to when you need dental care?		
Hospital	2	8.33
Health Center	16	66.67
University	1	4.17
Didn’t know	5	20.83
Should dental treatment for people with ASD happen in the same environment as people without ASD?		
Yes	19	79.17
No	2	8.33
Didn’t know	3	12.50

**Table 5 ijerph-21-00555-t005:** Conduct of dental professionals towards people with ASD, 2019–2020.

Variables	n	%
Differentiated attention for people with ASD about other patients with special needs		
Yes	5	35.71
No	8	57.14
Didn’t know	1	7.14
Differentiated appointment for people with ASD		
Yes	5	35.71
No	8	57.14
Didn’t know	1	7.14
Specific training in caring for people with ASD		
Yes	5	35.71
No	8	57.14
Didn’t know	1	7.14
Places reserved for people with ASD		
Yes	1	7.14
No	13	92.86
Oral health program for people with ASD		
Yes	1	7.14
No	12	85.71
Didn’t know	1	7.14
Qualified to use Nitrous Oxide		
Yes	4	28.57
No	10	71.43
A surgical center available where people with ASD are treated		
Yes	3	21.43
No	11	78.57
Time to First Appointment for Person with ASD		
Up to 01 week	6	46.15
Between 15 and 30 days	2	15.38
More than 30 days	1	7.69
Didn’t know	4	30.77
Guidance on oral health for those responsible for people with ASD		
Yes	14	100.00

## Data Availability

The data presented in this study are available on request from the corresponding author due to ethical reasons.

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
