# Peer review of "Public Dental Service Access Policies for People with Autism Spectrum Disorder (ASD) in Salvador, Bahia, Brazil: A Pre-Evaluation Study"

_ijerph, 2024, doi:10.3390/ijerph21050555_

Round 1
Reviewer 1 Report
Comments and Suggestions for Authors
The authors have engaged to understand the access policy of people with ASD through the mixed method approach, though this is not a novel study, but all population needs to be analyzed based on regional policies and decision implementation. Authors have made a great attempt in bringing out this aspect in this respective paper.
Q1. why was a pilot study not part of study /focus group discussions held to improvise on aspects of questionnaire?
Author Response
|
Response to Reviewer 1 Comments
|
||
|
1. Summary |
|
|
|
Thank you very much for taking the time to review this manuscript. Please find the detailed responses below and the corresponding revisions/corrections highlighted/in track changes in the re-submitted files
|
||
|
2. Questions for General Evaluation |
Reviewer’s Evaluation |
Response and Revisions |
|
Does the introduction provide sufficient background and include all relevant references? |
Yes |
|
|
Are all the cited references relevant to the research?
|
Yes |
|
|
Is the research design appropriate? |
Can be improved |
The research is an evaluability study, a kind of pre-evaluation. We explained its design in the text. (p. 2, paragraph 6, lines 88-90 and also in p.3, paragraph 1, lines 97-100) |
|
Are the methods adequately described? |
Yes |
|
|
Are the results clearly presented? |
Can be improved |
We reviewed the results, including the topic “Modeling of the policy” and improving the topics “The policy on access to dental services for people with ASD: clarifying its objective” and “Ways for overcoming weakness and dimensions and subdimensions for evaluation” |
|
Are the conclusions supported by the results?
|
Yes |
|
|
3. Point-by-point response to Comments and Suggestions for Authors |
||
|
Comments 1: The authors have engaged to understand the access policy of people with ASD through the mixed method approach, though this is not a novel study, but all population needs to be analyzed based on regional policies and decision implementation. Authors have made a great attempt in bringing out this aspect in this respective paper. Q1. why was a pilot study not part of study /focus group discussions held to improvise on aspects of questionnaire?
|
||
|
Response 1: Thank you for your question. We have done a pre-evaluation study, which was held together with a pilot study of an evaluation protocol upon access of people with ASD to dental health services administered at two Children and Youth Psychosocial Care Centers in Brazil. Therefore, the results of this pilot study contributed to the evaluability analysis and so we have clarified this on methods. (p. 2, paragraph 7, lines 93-96).
|
||

Reviewer 2 Report
Comments and Suggestions for Authors
Generally well written.
Authors may consider including information on how the interviewees are selected, and the sampling method/response rate for those who did the questionnaire, and whether there's any bias in parameters between those who responded, versus those who did not.
Author Response
|
Response to Reviewer 2 Comments
|
||
|
1. Summary |
|
|
|
Thank you very much for taking the time to review this manuscript. Please find the detailed responses below and the corresponding revisions/corrections highlighted/in track changes in the re-submitted files
|
||
|
2. Questions for General Evaluation |
Reviewer’s Evaluation |
Response and Revisions |
|
Does the introduction provide sufficient background and include all relevant references? |
Yes |
|
|
Are all the cited references relevant to the research?
|
Yes |
|
|
Is the research design appropriate? |
Yes |
|
|
Are the methods adequately described? |
Can be improved |
We reformulated “The Material and Methods” section according to reviewers' comments (p. 2-3, lines 87-131). |
|
Are the results clearly presented? |
Yes |
|
|
Are the conclusions supported by the results?
|
Yes |
|
|
3. Point-by-point response to Comments and Suggestions for Authors |
||
|
Comments 1: Generally well written. Authors may consider including information on how the interviewees are selected, and the sampling method/response rate for those who did the questionnaire, and whether there's any bias in parameters between those who responded, versus those who did not.
|
||
|
Response 1: Thank you for pointing this out. We agree that it is essential information that was missed. So, we have explained in the text that the interviewees were selected intentionally, and the questionnaires were part of a pilot study, the results of which were included to contribute to the evaluability analysis. |
||

Reviewer 3 Report
Comments and Suggestions for Authors
In the title
It is suggested to specify the type of study that was carried out: clinical trial, cross-sectional, longitudinal study...
Also, the place where the study was conducted or the population analyzed should be specified, to the extent possible. Indicating "in a Northeast Brazilian Capital" leaves any capital city as a choice.
Evaluability is defined as the degree to which a public intervention (policy, plan, program, standard) can be reliably and credibly evaluated.
Another definition is based on how the particular design features of a program or plan affect its ability to provide effective evaluation.
Clearly the word "evaluability" in the context used by the authors is somewhat misleading, as the authors focus on highlighting the current limitations to adequate dental care for patients with autism, and not on defining whether the Public Dental Services Access Policy can be reliably and credibly evaluated.
in the summary
The style recommendation of the journal should be followed, placing pertinent and noteworthy information. this section should be improved.
In the key words section
Authors are suggested to review the manuscript submission guidelines of the journal. It is noted the lack of willingness and interest of the authors to develop this part.
In this sense, we aimed to conduct a systematic and preliminary examination of this policy to define its objectives better and identify critical areas to be prioritized in the evaluation.
in the introduction
It specifically addresses the limitations and difficulties that the patient with ausitms faces in order to receive medical care and in this case dental care, on the one hand due to the condition of the signs and symptoms and the natural history of the disease, and on the other hand due to the "limited" training of the service providers in the entire chain of care.
However, little is said about the characteristics of The Dental Specialty Centers and the programs currently in place to provide care to this type of population. It is mentioned that there are documents and protocols of care but it is not specified what they are, it is not possible to access them to be able to visualize the dimension of care and the deficiencies they may have been not clarified.
Furthermore, the general objective is not adequately established, much less the specific ones, since the document shows the following:
1 “In this sense, we aimed to con-13 duct a systematic and preliminary examination of this policy to define its objectives better and identify critical areas to be prioritized in the evaluation.”
2 “The aim was to identify the clarity and adequacy of the documents' objectives and goals and propose ways of overcoming weaknesses and suggestions for additions. Finally, a proposal of dimensions and sub-dimensions was drawn up for evaluation.” This part was found in materials and methods.
3 “This article sought to carry out a systematic and preliminary examination of the policy on access to public dental services for people with ASD in a capital city in northeastern Brazil.”
4 “This type of study seeks to understand the different conceptions of managers and professionals responsible for implementing a policy or program about its objectives and to identify the weakest aspects that require in-depth evaluation” In this last paragraph it is specified that it is about evaluating people and their performance, and not specifically the health care program.
Not to mention, materials and methods is a section intended only to describe materials and methods, not to justify the study.
Also, in materials and methods
The type of study conducted is not specified.
The locality or municipality studied is not specified.
It is not specified why information was taken from November 2019 to February 2020.
There is no description of the questionnaire used, whether it was a validated and standardized instrument; the way in which the participants were selected in general is not established. Nor is it mentioned why the total number of questionnaires and interviews was determined, nor why there were fewer questionnaires for dental health professionals than for other participants. There is also no mention of the people who conducted the interviews, whether they were trained or whether they were suitable for the purpose.
The results, discussion and conclusions are conditioned to the modifications made to these sections.
In the discussion section it is noted
"Authors should discuss the results and how they can be interpreted from the perspective of previous studies and of the working hypotheses. The findings and their implications should be discussed in the broadest context possible. Future research directions may also be highlighted.." Again, there is a lack of understanding of the manuscript review process, and lack of attention by the author(s).
In the document and some tables there are abbreviations that are not fully described, the tables have information that conditions the exclusion of data. For example, missing or undeclared information.
It is suggested to review the document thoroughly, determine the general objective, identify and manage the pertinent information of each block of the manuscript, specify the materials and methods and write the conclusions based on the results obtained and allowing to cover the general objective of the manuscript.
Comments on the Quality of English LanguageSome paragraphs should be improved in their wording, it is not clear what the authors intend to report.
The meaning of what they intend to describe and inform should be revised.
Author Response
|
Response to Reviewer 3 Comments
|
||
|
1. Summary |
|
|
|
Thank you very much for taking the time to review this manuscript. Please find the detailed responses below and the corresponding revisions/corrections highlighted/in track changes in the re-submitted files.
|
||
|
2. Questions for General Evaluation |
Reviewer’s Evaluation |
Response and Revisions |
|
Does the introduction provide sufficient background and include all relevant references? |
Must be improved |
See point by point response |
|
Are all the cited references relevant to the research? |
Can be improved |
See point by point response |
|
Is the research design appropriate? |
Must be improved |
See point by point response |
|
Are the methods adequately described? |
Must be improved |
See point by point response |
|
Are the results clearly presented? |
Must be improved |
See point by point response |
|
Are the conclusions supported by the results? |
Must be improved |
See point by point response |
|
3. Point-by-point response to Comments and Suggestions for Authors |
||
|
Comments 1: In the title It is suggested to specify the type of study that was carried out: clinical trial, cross-sectional, longitudinal study... Also, the place where the study was conducted or the population analyzed should be specified, to the extent possible. Indicating "in a Northeast Brazilian Capital" leaves any capital city as a choice. Evaluability is defined as the degree to which a public intervention (policy, plan, program, standard) can be reliably and credibly evaluated. Another definition is based on how the particular design features of a program or plan affect its ability to provide effective evaluation. Clearly the word "evaluability" in the context used by the authors is somewhat misleading, as the authors focus on highlighting the current limitations to adequate dental care for patients with autism, and not on defining whether the Public Dental Services Access Policy can be reliably and credibly evaluated.
|
||
|
Response 1: Thank you for pointing this out. We agree in part with this comment. Therefore, we have adjusted the title to “Public Dental Services Access Policy for People with Autism Spectrum Disorder (ASD) in Salvador, Bahia, Brazil: a pre-evaluation study.” We highlighted the city where the study took place and specified the type of study (pre-evaluation study or evaluability study). As it was actually an evaluability study, we tried clarifying it in the methods and results.
|
||
|
Comments 2: in the summary The style recommendation of the journal should be followed, placing pertinent and noteworthy information. this section should be improved.
|
||
|
Response 2: Agree. We have, accordingly, reformulated the abstract emphasizing the methods, results and conclusions.
|
||
|
Comments 3: In the key words section Authors are suggested to review the manuscript submission guidelines of the journal. It is noted the lack of willingness and interest of the authors to develop this part. In this sense, we aimed to conduct a systematic and preliminary examination of this policy to define its objectives better and identify critical areas to be prioritized in the evaluation. Response 3: We apologize for the lack of keywords. It was a mistake, not a lack of willingness and interest in developing this part. We defined 5 key words.
|
||
|
Comments 4: in the introduction It specifically addresses the limitations and difficulties that the patient with ausitms faces in order to receive medical care and in this case dental care, on the one hand due to the condition of the signs and symptoms and the natural history of the disease, and on the other hand due to the "limited" training of the service providers in the entire chain of care. However, little is said about the characteristics of The Dental Specialty Centers and the programs currently in place to provide care to this type of population. It is mentioned that there are documents and protocols of care but it is not specified what they are, it is not possible to access them to be able to visualize the dimension of care and the deficiencies they may have been not clarified. Response 4: Agree. We have, accordingly, described the characteristics of the Dental Specialty Centers (p. 2, paragraph 3, lines 60-67). We also include a paragraph on the current policies and programs to provide care to this population in Brazil (p. 2, paragraph 4, lines 68-77), although, as it is the objective of the evaluability study, the policy is more detailed in the results. |
||
|
|
||
|
Comments 5: Furthermore, the general objective is not adequately established, much less the specific ones, since the document shows the following: 1 “In this sense, we aimed to con-13 duct a systematic and preliminary examination of this policy to define its objectives better and identify critical areas to be prioritized in the evaluation.” 2 “The aim was to identify the clarity and adequacy of the documents' objectives and goals and propose ways of overcoming weaknesses and suggestions for additions. Finally, a proposal of dimensions and sub-dimensions was drawn up for evaluation.” This part was found in materials and methods. 3 “This article sought to carry out a systematic and preliminary examination of the policy on access to public dental services for people with ASD in a capital city in northeastern Brazil.” 4 “This type of study seeks to understand the different conceptions of managers and professionals responsible for implementing a policy or program about its objectives and to identify the weakest aspects that require in-depth evaluation” In this last paragraph it is specified that it is about evaluating people and their performance, and not specifically the health care program. Not to mention, materials and methods is a section intended only to describe materials and methods, not to justify the study. Response 5: Thank you for pointing this out. We agree that the objectives were written in different forms in diverse places. We have, accordingly, clarified it in the introduction (p. 2, paragraph 5, lines 81-85).
|
||
|
Comments 6: Also, in materials and methods The type of study conducted is not specified. The locality or municipality studied is not specified. It is not specified why information was taken from November 2019 to February 2020. There is no description of the questionnaire used, whether it was a validated and standardized instrument; the way in which the participants were selected in general is not established. Nor is it mentioned why the total number of questionnaires and interviews was determined, nor why there were fewer questionnaires for dental health professionals than for other participants. There is also no mention of the people who conducted the interviews, whether they were trained or whether they were suitable for the purpose. Response 6: Thank you for the comments. We have highlighted that we have carried out an evaluability study or pre-evaluation study (p. 2, paragraph 6, lines 87-90) and also the steps of the study were detailed in the “Material and Methods” (p. 2, paragraph 7, lines 91-96 and p. 3, paragraph 1, lines 97-100). The municipality studied was specified (p. 2, paragraph 6, line 88), it was clarified why the information was taken from December 2019 to February 2020 (p. 3, paragraph 3, lines 103-105), a brief description of the objectives of the questionnaire used in the pilot study was included (p. 4, paragraph 1, lines 123-127), we also clarify that the sample was selected intentionally (p. 2, paragraph 7, line 93), that there were no refusals to questionnaires or interviews application (p. 3, paragraph 3, lines 104-105). The number of respondents to questionnaires for each category depends on who was in the CAPSi on the day of the pilot study. Information about the person who conducted the interviews was also included (p. 3, paragraph 4, lines 109-110). |
||
|
Comments 7: The results, discussion and conclusions are conditioned to the modifications made to these sections. In the discussion section it is noted "Authors should discuss the results and how they can be interpreted from the perspective of previous studies and of the working hypotheses. The findings and their implications should be discussed in the broadest context possible. Future research directions may also be highlighted.." Again, there is a lack of understanding of the manuscript review process, and lack of attention by the author(s). Response 7: The results were adjusted to the significant reviews indicated by the reviewer. Then, we discuss the findings and their implications for the policy implementation in the city under study. No evaluability studies on this purpose were found in the literature, nor articles about specific public oral health policies for people with ASD, limiting the discussion context. Though attempting to “the broadest context possible,” we discussed the results with the existent literature about oral health care for ASD people and the organization of public oral health services.
|
||
|
Comments 8: In the document and some tables there are abbreviations that are not fully described, the tables have information that conditions the exclusion of data. For example, missing or undeclared information. It is suggested to review the document thoroughly, determine the general objective, identify and manage the pertinent information of each block of the manuscript, specify the materials and methods and write the conclusions based on the results obtained and allowing to cover the general objective of the manuscript. Response 8: We reviewed the abbreviations throughout the text and tables. As suggested, the document was examined thoroughly, considering the general objective and the pertinent information of each block of the manuscript, specifying the materials and methods, and the discussion and conclusions were rewritten based on the results obtained. Thank you very much for this careful review.
|
||
|
4. Response to Comments on the Quality of English Language |
||
|
Point 1: Some paragraphs should be improved in their wording, it is not clear what the authors intend to report. The meaning of what they intend to describe and inform should be revised.
|
||
|
Response 1: We agree that some paragraphs were not clear, but we are not sure they are the same as the reviewer suggested because they were not indicated. All the text was reviewed by two fluent English-speaking colleagues.
|
||

Round 2
Reviewer 3 Report
Comments and Suggestions for Authors
After reviewing the manuscript in this round of revision, changes, improvements and adequate structure are observed that allow it to be considered for publication.
I thank the authors for taking the time to make these changes, which allow us to have a manuscript based on the scientific method, the pillar of research.
Author Response
Once more, we would like to thank the reviewer for all the suggestions in Round 1.
Reviewing the aspects pointed out indeed qualified the paper.